

# The effects of COVID-19 on semen parameter values in healthy males: a single-centre, retrospective study

Anil Erdik[1], Asli Merve Gokce[2] and Ahmet Gokce[3]

[1] Department of Urology, Karasu State Hospital, Sakarya, Turkey
[2] İstanbul Medeniyet University, Istanbul, Turkey
[3] Department of Urology, Sakarya University, Sakarya, Turkey

## ABSTRACT

**Background.** To investigate the effects of the novel coronavirus disease 2019 (COVID-19) on spermatogenesis and the potential impact on patients with normal semen quality before a COVID-19 diagnosis.

**Methods.** This retrospective study included 22 male patients (aged 18–48 years) diagnosed with mild COVID-19 *via* reverse transcription polymerase chain reaction RT-PCR using combined oropharyngeal and nasopharyngeal swabs between April 2020 and June 2021. All participants had prior normal semen parameters (World Health Organization (WHO) standards) confirmed at our male infertility outpatient clinic before COVID-19 infection. Post-COVID-19 semen analyses were performed three months after diagnosis to evaluate subacute effects. Exclusion criteria included hospitalization for COVID-19, pre-existing abnormal semen parameters, or history of testicular surgery.

**Results.** The mean age of the patients was $31.8 \pm 5.9$ years. An abnormality was detected in at least one parameter value in the post-COVID-19 semen analysis in nine patients (40.9%) whose semen quality was normal before COVID-19. When post-COVID-19 semen samples of the patients were divided into normal and abnormal groups, total sperm motility, progressive motility, and normal morphology were found to be significantly decreased, immotility significantly was increased, and semen pH tended to be more alkaline in the abnormal group.

**Conclusion.** Even though the effects of COVID-19 on spermatogenesis are not fully understood, COVID-19 infection may have negative effects on semen quality and impair fertilization.

Subjects Urology, COVID-19
Keywords Male infertility, Normospermia, SARS-CoV-2, Semen analyses

# INTRODUCTION

The global outbreak of the novel coronavirus disease 2019 (COVID-19), caused by the novel severe acute respiratory syndrome coronavirus 2 (SARS-CoV-2), was first identified in Wuhan, China and rapidly spread worldwide, leading to significant public health challenges (*Zhu et al., 2020*). SARS-CoV-2 was defined as the viral pathogen for the pandemic (*Zhou et al., 2020*) with two main modes of transmission: respiratory droplets and close contact (*Chen et al., 2020*) (Portions of this text were previously published as part of a preprint

Corresponding author
Anil Erdik, anilerdik@gmail.com

(*Erdik, Gokce & Gokce, 2023*). The hallmark of COVID-19 is its severe impact on the respiratory system, which can progress to multiorgan failure (*Sun et al., 2020*). A study conducted by *Sun et al. (2020)* detected SARS-CoV-2 viral RNA in a wide range of organs in affected patients.

The entry of SARS-CoV-2 into host cells involves two key proteins: angiotensin-converting enzyme 2 (ACE2), which serves as the viral receptor, and transmembrane serine protease 2 (TMPRSS2), a protease that cleaves the spike protein to activate fusion (*Hoffmann et al., 2020*). When TMPRSS2 is absent or minimally expressed, viral entry shifts to the endosomal pathway, and relies on ACE2 and lysosomal proteases such as cathepsins (*Lamers & Haagmans, 2022*). The systemic expression of ACE2 extends beyond the lungs to include the digestive, circulatory, central nervous, and urogenital systems, alongside reproductive organs, creating pathways for direct viral invasion by SARS-CoV-2 (*Zhang et al., 2020*). The entry of SARS-CoV-2 into cells is facilitated by its high affinity for ACE2, which serves as the principal receptor for viral attachment and internalization (*Stanley et al., 2020*; *Colaco et al., 2021*).

Although SARS-CoV-2 primarily induces severe pulmonary damage, its systemic effects—particularly the cytokine storm observed in severe COVID-19—can precipitate multi-organ failure (*Nazerian et al., 2022*). The male reproductive tract (MRT) is not spared, yet the mechanisms remain debated. While some studies using single-cell RNA sequencing datasets identified high ACE2 and TMPRSS2 expression in testicular cells (*e.g.*, spermatogonia, somatic cells) (*Qi et al., 2021*), others found no evidence of co-expression, making an argument against direct viral invasion (*Stanley et al., 2020*). This discrepancy implies that testicular impairment may result from indirect pathways, such as inflammatory cascades or fever-induced stress, rather than ACE2-mediated viral entry (*Rago & Perri, 2023*).

Current evidence suggests that systemic inflammation during SARS-CoV-2 infection may compromise the blood-testis barrier (BTB), enabling viral access to the germinal compartment (*Dabizzi, Maggi & Torcia, 2024*). Similar to other viruses such as H1N1, Zika, Ebola, HIV, hepatitis, and HPV, SARS-CoV-2 has been implicated in testicular dysfunction through mechanisms such as oxidative-inflammatory damage, atrophy of seminiferous tubules and Sertoli cells, and diminished Leydig cell mass, culminating in hypotestosteronemia (*Akhigbe et al., 2022*). Furthermore, high fever and cytokine-driven inflammation, marked by leukocyte infiltration, exacerbate germ cell apoptosis, impair spermatogenesis, and degrade seminal parameters (*Xu et al., 2000*).

Sex disparity is a well-documented risk factor for severe COVID-19 outcomes, attributed to mechanisms such as sex hormone-mediated modulation of viral entry, immune response, and coagulation pathways (*Pivonello et al., 2021*). Men exhibit a less robust immune response, leading to heightened disease severity and mortality, often driven by dysfunctional inflammation, disseminated coagulopathy, and thromboembolism (*Giannis, Ioannis & Gianni, 2020*). SARS-CoV-2 infection of endothelial cells triggers inflammatory infiltration and apoptosis, resulting in endotheliopathy, platelet activation, and widespread thrombosis, which manifests as microvascular injury, ischemic stroke, or acute coronary events (*Jackson, Darbousset & Schoenwaelder, 2019*; *Beyrouti et al., 2020*; *Connors & Levy,*

*2020*; *Edler et al., 2020*; *Franchini et al., 2024*). Sex disparity is a well-documented risk factor for severe COVID-19 outcomes, with men exhibiting higher mortality rates. This necessitates proactive monitoring of recovered patients—particularly those with severe disease—to assess potential damage to extrapulmonary organ systems, including the MRT. While cardiovascular complications (*e.g.*, myocardial injury) require acute interventions like reperfusion therapy (*Cataldo et al., 2021*), persistent systemic inflammation may also impair MRT function, contributing to long-term reproductive sequelae. Severe COVID-19 cases, particularly those requiring intensive care, frequently exhibit multi-organ dysfunction, with emerging evidence also suggesting reproductive tissue involvement (*Gacci et al., 2021*). Proactive monitoring of recovered patients, especially those with severe cardiovascular or respiratory sequelae, is critical, as cytokine storms and oxidative stress may indirectly disrupt spermatogenesis and semen quality. SARS-CoV-2's systemic impact includes testicular involvement *via* ACE2-mediated entry or cytokine storms. The blood-testis barrier disruption enables viral access, paralleling mechanisms seen in H1N1/Zika (*Akhigbe et al., 2022*). This foundation contextualizes our focus on isolated semen parameter changes.

Despite growing evidence of SARS-CoV-2's impact on semen parameters (*e.g.*, reduced motility, increased DNA fragmentation), critical gaps remain. This study addresses these gaps through two primary aims; to examine the effects of COVID-19 on spermatogenesis, and allow understanding of the possible effects of COVID-19 infection on patients with previously normal semen parameter values (according to World Health Organization (WHO) standards).

## MATERIALS & METHODS

### Study design and ethical considerations

The study was designed as a retrospective analysis and carried out at a single center, the Sakarya University Training and Research Hospital. Ethical approval was obtained from the Institutional Review Board of the Sakarya University Ethics Committee (approval number: 71522473-050.01.04-186713-303). The need for informed consent was waived due to the retrospective nature of the study. All procedures were performed in line with the ethical standards outlined in the 1964 Declaration of Helsinki and its later amendments.

### Data collection period

Data collection was conducted retrospectively between April 2020 and June 2021, and included patients diagnosed with COVID-19 who were followed up with at the urology outpatient clinic. This period allowed for the inclusion of sufficient cases and ensured that semen analyses were performed three months post-diagnosis to evaluate the subacute effects of COVID-19. The timeframe aligns with the clinical management and follow-up protocols implemented during the pandemic.

### Patient eligibility and exclusion criteria

This study enrolled 222 male patients (aged 18–48 years) diagnosed with mild COVID-19 *via* reverse transcription polymerase chain reaction (RT-PCR) using combined

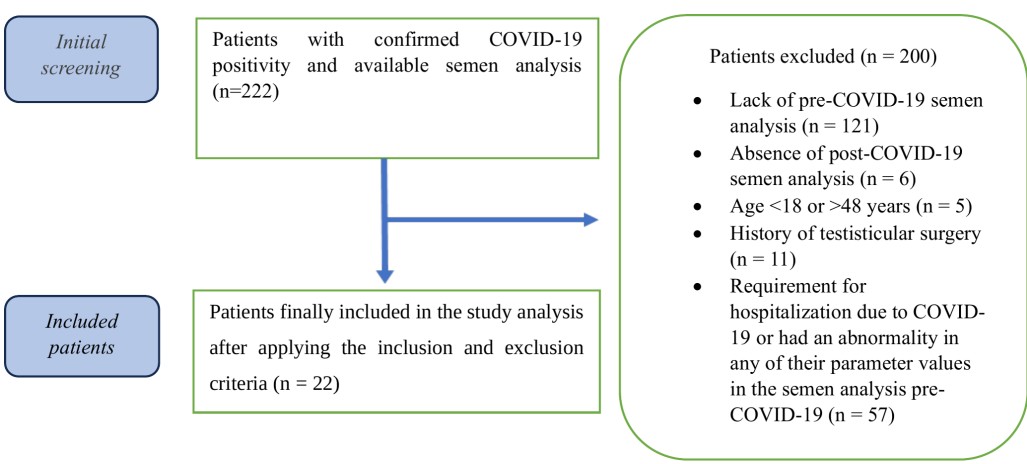

**Figure 1 Flowchart of the patient selection and grouping process in this study.**

oropharyngeal and nasopharyngeal swabs between April 2020 and June 2021. Exclusion criteria were applied as follows:

1. lack of pre-COVID-19 semen analysis ($n = 121$),
2. absence of post-COVID-19 semen analysis ($n = 6$),
3. age <18 or >48 years ($n = 5$),
4. history of testicular surgery ($n = 11$),
5. requirement for hospitalization due to COVID-19 or had an abnormality in any of their parameter values in the semen analysis pre-COVID-19 ($n = 57$).

From an initial cohort of 222 patients presenting with infertility complaints, 200 were excluded based on the above criteria. Ultimately, 22 patients with confirmed normal pre-COVID-19 semen parameters (WHO standards) were included in the final analysis (see Fig. 1 for the patient selection flowchart).

## Semen analysis and COVID-19 treatment protocol

Semen analysis was performed according to WHO 2010 guidelines in our hospital's andrology laboratory (*World Health Organization, n.d.*). Samples were collected through masturbation and analyzed within 30 min of liquefaction at room temperature. Internal and external quality controls of the laboratory were routinely performed.

COVID-19 treatment for all patients included oral favipiravir tablets for five days, social isolation, and enoxaparin sodium for thromboembolism prophylaxis. Antibiotics were not prescribed as no bacterial infections were detected, and acetaminophen was used symptomatically for fever and pain. No antioxidants were taken in any of the patients in our study.

## Follow-up time

The follow-up period in this study was designed to capture the acute effects of COVID-19 on semen parameters, with analyses conducted three months post-diagnosis. This timeframe encompassed one full spermatogenesis cycle and allowed for the evaluation of subacute

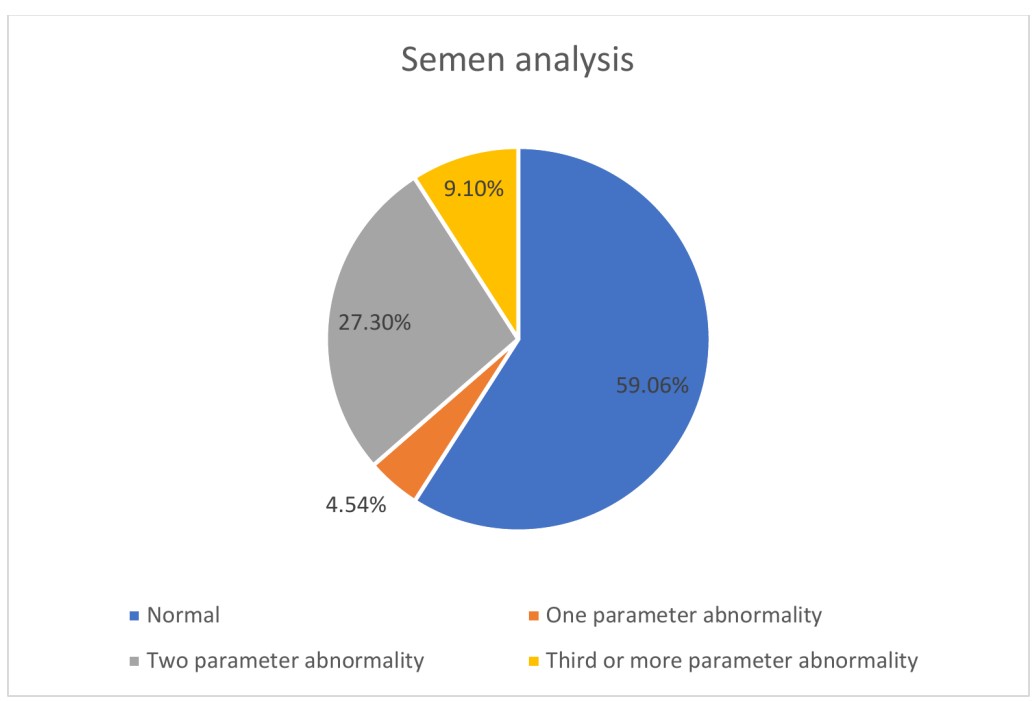

**Figure 2** Evaluation of semen analysis before and after COVID-19.

changes. Future investigations will incorporate longer follow-up durations to assess the long-term impacts on reproductive health.

## Statistical analysis

Statistical analyses were conducted with SPSS (version 23.0.0, Chicago, IL, USA). Frequencies are presented as absolute numbers and percentages. We assessed the distribution of the data using the Shapiro–Wilk test. In case the normality criteria were not met, the Mann Whitney $U$ test was used for continuous variables. Continuous data are presented as median with interquartile range (IQR).

## RESULTS

The mean age of the patients was $31.8 \pm 5.9$ years. Twenty-two men with mild COVID-19 were included, none of whom required hospitalization or reported testicular pain during follow-up. Post-COVID-19 semen analysis revealed abnormalities in at least one parameter (viscosity, liquefaction time, progressive motility, or morphology) in nine patients (40.9%) who had normal pre-infection semen quality (Fig. 2). One patient (4.5%) exhibited one abnormal parameter, six patients (27.3%) showed abnormalities in two parameters, and two patients (9.1%) had three or more abnormal parameters (Fig. 3).

Patients' post-COVID-19 sperm samples were categorized as normal ($n = 13$) or abnormal ($n = 9$). The abnormal group demonstrated significant reductions in total motility, progressive motility, and morphologically normal sperm, as well as increased

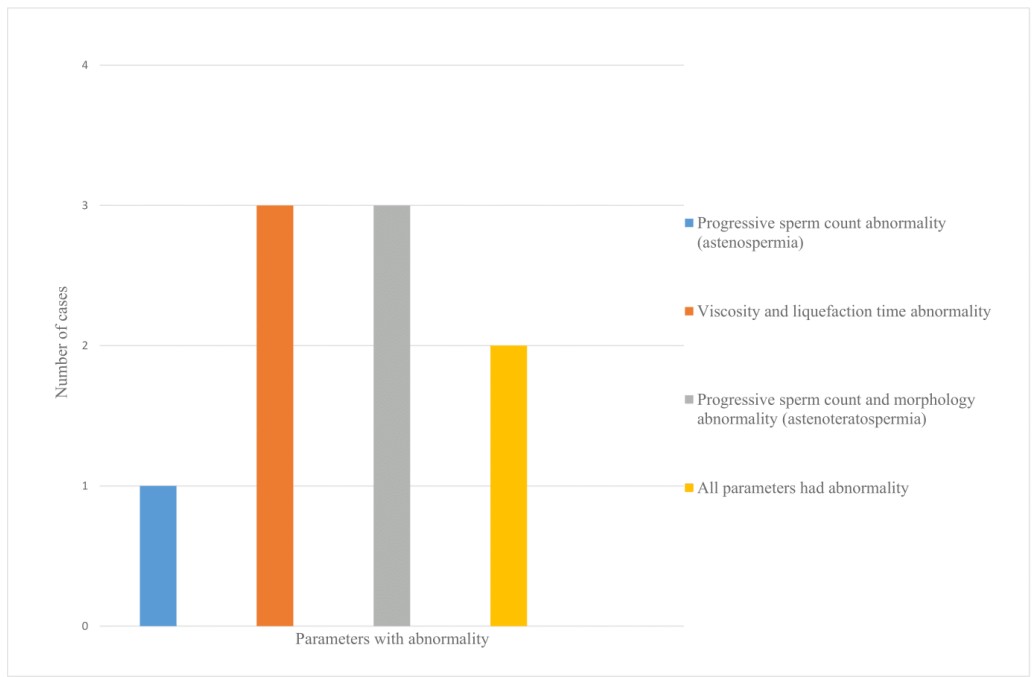

**Figure 3** **Distribution of patients with abnormal semen analysis parameters after COVID-19.**

immotility and a trend toward alkaline semen pH ($p < 0.05$; Table 1). In contrast, median sperm concentration did not differ significantly between groups.

## DISCUSSION

Since the onset of the COVID-19 pandemic in December 2019, substantial progress has been made in understanding the disease's progression, its impact on various organ systems, and strategies to limit its transmission. Investigating the long-term effects on different physiological systems, including the MRT, remains a key concern for clinicians (*Ding et al., 2020*). The testes are primarily composed of seminiferous tubules and interstitial tissue. Sperm production takes place within the seminiferous tubules, supported by spermatogonia (the germ cells responsible for spermatogenesis) and Sertoli cells, which provide structural and nutritional support.

ACE2 receptors are present in several testicular districts, including germinal, Sertoli, and Leydig cells (*Ma et al., 2021b*). Additionally, ACE2 receptors are expressed in the epididymis epithelium and specific zones of the seminal vesicles (*Rago & Perri, 2023*). TMPRSS2 is highly expressed in both germinal cells and prostatic tissue (*Dabizzi, Maggi & Torcia, 2024*). The testes is a potential target for SARS-CoV-2. The effect of testosterone on the male reproductive system and the presence of the androgen receptor supporting the TMPRSS2 gene likely further promote viral uptake (*Wang & Xu, 2020*; *Wambier & Goren, 2020*). Angiotensin 2 also affects fertilization and sperm motility by stimulating angiotensin 2 type 1 receptor (AT1R) and type 2 receptor (AT2R) (*Gianzo et al., 2016*; *Aitken, 2021a*). Prolonged high levels of angiotensin 2 can result in apoptosis and cellular

**Table 1  Comparison of semen parameters between groups according to post-COVID-19.**

| Semen parameters | Normal group (n = 13) | Abnormal group (n = 9) | p value |
|---|---|---|---|
| Semen volume (mL)[*] | 3.00 (1.40) | 3.50 (0.75) | 0.209[a] |
| Semen pH[*] | 7.50 (0.05) | 7.60 (0.15) | 0.043[a] |
| Sperm concentration (millions/mL)[*] | 54.00 (35.00) | 58.00 (45.00) | 0.744[a] |
| Total sperm number (millions)[*] | 156.00 (85.85) | 197.20 (208.85) | 0.601[a] |
| Motility [#] | 59.46 (12.94) | 45.00 (12.59) | 0.001[a] |
| Total motile sperm concentration (millions/mL)[*] | 30.00 (22.50) | 24.00 (29.50) | 0.209[a] |
| Progressive sperm concentration (millions/mL)[*] | 24.00 (21.50) | 16.00 (25.00) | 0.060[a] |
| Nonprogressive sperm concentration (millions/mL)[*] | 5.00 (1.50) | 6.00 (5.50) | 0.292[a] |
| Immotile sperm count (millions/mL)[*] | 24.00 (12.00) | 30.00 (27.00) | 0.209[a] |
| Progressive motility[#] | 50.00 (14.08) | 30.00 (19.06) | <0.001[a] |
| Nonprogressive motility[#] | 8.57 (3.78) | 13.33 (7.23) | 0.082[a] |
| Immotility[#] | 40.54 (12.94) | 55.00 (12.59) | 0.001[a] |
| Normal morphology[#] | 5.00 (1.00) | 3.0 (3.00) | 0.030[a] |

Notes.
[*]Median (IQR).
[#]n(%).
[a]Mann–Whitney $U$ test.

senescence of spermatozoa. Additionally, inflammatory cytokines and oxidative stress caused by SARS-CoV-2 can impair the function of Leydig cells, reducing testosterone synthesis, and negatively influence germinal cells, hindering the process of spermatogenesis (*Rago & Perri, 2023*).

Emerging evidence indicates that SARS-CoV-2 can directly interact with sperm cells and potentially lead to male fertility issues, as highlighted in a recent study (*Aitken, 2021a*). These cells are likely to have the complete repertoire of receptors needed to support angiotensin signaling pathways, including ACE2, the cellular entry point of the coronavirus. Thus, the angiotensin system may play an essential role in sustaining sperm viability and function, and contact with the virus will have adverse effects on the reproductive systems of infected males (*Aitken, 2021a*). Current data on the impact of SARS-CoV-2 infection on testicular function are primarily sourced from a few autopsy studies of testis and epididymis samples from deceased COVID-19 patients. Despite conflicting evidence on the presence of the virus in the testes, there has been consistent documentation of both macroscopic and microscopic testicular damage (*Li et al., 2020*; *Ma et al., 2021b*). *Li et al. (2020)* were an early group that examined the testicular and epididymal autopsies of patients who died from COVID-19. They detected interstitial edema, congestion, and red blood cell exudation in these organs. It has been documented that the seminiferous tubules of patients who died from COVID-19 exhibit a significantly greater concentration of apoptotic cells than those in control cases. *Li et al. (2020)* also detected oligozoospermia, leukospermia, and elevated interleukin-6 (IL-6) levels in the semen of hospitalized COVID-19 patients. In another investigation, *Holtmann et al. (2020)* stratified patients into mild and moderate categories according to COVID-19 symptom severity. The study analyzed the influence of symptom severity on
semen parameters by comparing infected men to a control group (*Holtmann et al., 2020*). The authors found that men with moderate symptoms had a decreased total sperm count and impaired motility, whereas those with mild symptoms showed no notable changes in semen quality. Additionally, when participants were grouped by fever status, semen parameters were lower in the febrile group but remained within normal limits (*Holtmann et al., 2020*). *Ruan et al. (2021)* compared COVID-19 patients with a control group. Sperm concentration, total sperm count, and total motility were significantly decreased in the group with COVID-19, but the values were within the normal range. Different COVID-19 symptoms also made no significant difference on semen parameter values. The paper noted that semen parameter values may deteriorate in patients with a longer recovery period (*Ruan et al., 2021*). These studies suffer from a lack of pre-COVID-19 data as a reference point and were instead based on comparisons with healthy controls (*Aitken, 2021b*). Other studies concluded that there is a significant reduction in total sperm count and total motility that may be associated with direct sperm harm mediated by SARS-CoV-2 (*Ma et al., 2021a*; *Guo et al., 2021*). A previous study documented that COVID-19 has a significant impact on many semen parameters, including semen volume, total sperm motility, percentage of forward motile spermatozoa, and normal sperm morphology (*Koç & Buğra, 2021*). Similarly, in a multicenter study, Erbay and colleagues (*2021*) grouped patients as mild and moderately symptomatic and compared semen analyses of COVID-19 patients at least three months after recovery with those of a pre-COVID-19 period. They found that while sperm vitality, progressive motility, and total motility were lower in the mildly symptomatic group than the pre-COVID-19 levels, all sperm parameters were negatively affected by the disease in the moderately symptomatic group (*Erbay et al., 2021*). Cases reported from an *in vitro* fertilization clinic in India had pre- and post-COVID-19 semen analysis data. Patients were normozoospermic prior to COVID-19, but semen analyses performed one month after COVID-19 infection showed a dramatic decrease in sperm count, motility, morphology, and DNA integrity. The most surprising aspect of the data was that sperm DNA and morphology continued to deteriorate in the fourth month post-COVID-19 infection, and sperm count and motility did not reach pre-COVID-19 levels (*Mannur et al., 2021*). Although publications have stated that sperm quality is significantly impaired in patients with COVID-19 infection (*Pazir et al., 2021*), it is still controversial whether this effect is directly mediated by the viral infection of the male reproductive organs (*Dabizzi, Maggi & Torcia, 2024*).

The primary objective of this study was to evaluate changes in semen parameters before and after COVID-19 infection by focusing on patients with previously normal semen quality. To increase the objectivity of the comparison, patients with normal pre-COVID-19 semen parameter values were included in our study. Unlike prior studies using heterogeneous cohorts, our exclusion of confounders (*e.g.*, pre-existing subfertility) isolates COVID-19's temporal impact. This approach reveals 40.9% developed new semen abnormalities despite mild infection. One of the most noteworthy observations in this study was that 41% of participants with previously normal semen profiles exhibited abnormalities in their semen parameters following COVID-19 infection. When patients were categorized based on their post-infection semen analysis, those in the abnormal group demonstrated

elevated semen pH levels and a higher proportion of immotile sperm. Additionally, reductions were observed in total and progressive motility as well as normal morphological forms. Among these individuals, one was diagnosed with asthenozoospermia, while five exhibited asthenoteratozoospermia. Notably, there were no statistically significant differences in sperm concentration or total sperm count between the normal and abnormal groups. These findings are consistent with prior studies, suggesting a detrimental impact of COVID-19 on multiple aspects of semen quality (*Koç & Buğra, 2021*; *Erbay et al., 2021*; *Mannur et al., 2021*). This study uniquely demonstrates that even mild COVID-19 can disrupt semen parameters in previously normozoospermic men—a population seldom isolated in prior research.

To date, limited research has explored the relationship between COVID-19 infection and alterations in semen quality (*Aitken, 2021b*). A notable contribution by *Hajizadeh Maleki & Tartibian (2021)* reported significant reductions in semen volume, progressive motility, normal morphology, sperm count, and DNA integrity following infection. Furthermore, their findings demonstrated a link between diminished semen parameters and elevated markers of oxidative stress and inflammation (*Hajizadeh Maleki & Tartibian, 2021*). These observations are supported by additional studies indicating that oxidative stress plays a key role in the deterioration of semen quality post-COVID-19 infection (*Falahieh et al., 2021*). However, current evidence remains inconclusive regarding whether these changes stem directly from viral effects on spermatogenesis or are secondary to systemic inflammation and oxidative damage induced by the disease (*Aitken, 2021b*). Given the uncertainty surrounding the long-term impact on sperm quality, *Nassau et al. (2021)* recommended fertility assessments in men recovering from COVID-19 as a precautionary measure. While sperm DNA integrity was not assessed, oxidative mechanisms implicated in COVID-19 may explain our observed motility loss. *Hajizadeh Maleki & Tartibian (2021)* reported elevated DNA fragmentation post-infection, correlating with reduced fertility potential.

Our findings of post-COVID-19 semen abnormalities, particularly reduced sperm motility, align with emerging evidence of SARS-CoV-2's indirect effects on male reproductive health. Recent studies suggest that spermatozoa may trigger the formation of extracellular traps (ETs) in monocytes, which could inhibit sperm motility and compromise reproductive function (*Mamtimin et al., 2022*). During SARS-CoV-2 infection, the germinal epithelium is exposed to heightened levels of pro-inflammatory molecules and oxidative stress, affecting both immune cells (*e.g.*, leukocytes) and spermatozoa. Remarkably, these cells share a defensive mechanism termed ETosis, where nuclear DNA is released to entrap viral particles. In spermatozoa, structures like sperm extracellular traps (SETs) have been observed to cluster and neutralize SARS-CoV-2 particles within spermatids and mature sperm (*Hallak et al., 2024*). This mechanism may partially explain the impaired sperm motility observed in COVID-19 patients, as ETosis-driven DNA release could exacerbate sperm DNA fragmentation, a finding supported by studies reporting persistent DNA damage even after mild infections (*Erbay et al., 2021*). COVID-19-induced oxidative stress correlates with sperm DNA fragmentation, potentially impairing fertility (*Hajizadeh Maleki & Tartibian, 2021*). Although our study did not directly measure ETs or cell-free DNA (cfDNA), the post-COVID decline in semen quality likely reflects these

inflammatory and oxidative pathways. Future research should explore whether sperm ETs contribute to long-term DNA damage in recovered patients, potentially guiding therapies to mitigate post-viral infertility.

In a follow-up telephone interview conducted at a median of 44 months post-infection, 16 of the 22 patients (72.7%) reported achieving fatherhood. This self-reported outcome suggests that despite transient abnormalities observed in semen parameters following mild COVID-19 infection, long-term fertility may remain largely preserved in a majority of individuals. These findings align with previous studies, such as those by *Gacci et al. (2021)* and *Holtmann et al. (2020)*, which reported reversible impairments in semen quality post-COVID-19, particularly in patients with mild symptoms. Similarly, *Erbay et al. (2021)* noted recovery in semen parameters over time, especially in mildly affected individuals, although moderate cases exhibited more persistent deficits.

The relatively high paternity rate observed in our cohort may reflect several contributing factors: partial recovery of spermatogenesis over time, compensatory mechanisms in testicular function, or the limitations of semen analysis in fully predicting reproductive outcomes. It is also possible that mild disease severity in our patient group minimized long-term damage to the reproductive axis. Nevertheless, these interpretations should be approached cautiously. The study relied on self-reported fertility status, lacked comprehensive hormonal and oxidative stress assessments, and did not include partner fertility data or time-to-pregnancy metrics. Moreover, the small sample size and retrospective nature of the study limited its generalizability. Taken together, our findings support the hypothesis that COVID-19 may cause short-term disruptions in semen quality even in normozoospermic individuals, but these changes do not necessarily translate into permanent infertility. However, given the potential for variability based on disease severity, comorbidities, and individual susceptibility, long-term prospective studies with larger populations and detailed reproductive follow-up are needed to confirm these observations.

The single-center nature of this study ensured strict adherence to standardized protocols for semen analysis, RT-PCR testing, and clinical evaluations, and minimized inter-institutional variability. Additionally, the homogeneity of the cohort (*e.g.*, exclusion of patients with pre-existing fertility issues or comorbidities) allowed for a clearer examination of COVID-19's isolated effects on semen parameters, a feature often unattainable in multi-center studies with diverse populations. Despite its strengths, this study has several limitations. First, the single-center design, while ensuring standardized protocols and minimizing inter-observer variability, may have limited the generalizability of findings to broader populations. Second, the small sample size ($n = 22$) reduced the statistical power to detect subtle effects, particularly in subgroup analyses.Third, the short follow-up period (3 months post-infection) precluded assessment of long-term recovery or persistent infertility. Finally, DNA integrity rate data could not be evaluated because it could not be examined in our clinic until before the COVID-19 pandemic. Future multi-center studies with larger cohorts and extended follow-up durations are needed to validate these findings and explore mechanisms such as hormonal dysregulation or oxidative stress.

## CONCLUSION

By focusing on a rigorously selected cohort with pre-COVID normozoospermia, this study advances our understanding of SARS CoV-2's direct reproductive toxicity, independent of confounding factors. The observed decline in semen parameters, though reversible in most cases, suggest that COVID-19 may transiently impair male fertility, warranting clinical attention.

### Funding

The authors received no funding for this work.

### Competing Interests

The authors declare there are no competing interests.

### Author Contributions

- Anil Erdik performed the experiments, analyzed the data, prepared figures and/or tables, and approved the final draft.
- Asli Merve Gokce conceived and designed the experiments, authored or reviewed drafts of the article, and approved the final draft.
- Ahmet Gokce conceived and designed the experiments, authored or reviewed drafts of the article, and approved the final draft.

### Human Ethics

The following information was supplied relating to ethical approvals (i.e., approving body and any reference numbers):

The present study protocol was reviewed and approved by the Institutional Review Board of Sakarya University School of Medicine (Approval Number through 71522473-050.01.04-186713-303).

### Data Availability

The raw data are available in the Supplementary Files.

### Supplemental Information

Supplemental information for this article can be found online at http://dx.doi.org/10.7717/peerj.19864#supplemental-information.

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
