# Peer review of "The effects of COVID-19 on semen parameter values in healthy males: a single-centre, retrospective study"

_PeerJ, doi:10.7717/peerj.19864_

## Round 0.1 · original submission · Major Revisions

Dear authors,

Thank you for your submission. Please, address accordingly the concerns raised by the reviewers.

**Language Note:** The review process has identified that the English language must be improved. PeerJ can provide language editing services - please contact us at [email protected] for pricing (be sure to provide your manuscript number and title). Alternatively, you should make your own arrangements to improve the language quality and provide details in your response letter. – PeerJ Staff

Reviewer 1 ·

Basic reporting

The manuscript aims to investigate the impact of SARS-CoV-2 infection on spermatogenesis and its potential effects on patients who exhibited usual semen quality before their COVID-19 diagnosis. Although this topic has been explored in previous scientific literature with larger cohorts, this paper is well-structured and provides valuable insights for practitioners. However, it does contain a few minor grammatical errors. Therefore, I recommend the publication of this manuscript in the journal, provided that the authors address and correct some major issues before publication.

1) The manuscript contains several grammatical errors that need to be addressed. I encourage the authors to review the paper thoroughly to ensure grammatical accuracy.

2) The methods must be clarified in the abstract; currently, readers cannot easily comprehend the basic methodological outline of the study as it is presented.

3) In the Introduction section, the authors should enhance their description of the disease, highlighting its systemic nature and briefly explaining the fundamental mechanisms that render the male reproductive tract (MRT) vulnerable to SARS-CoV-2 infection and to establish a better standard for comparing the pathophysiology of different viral infections affecting the male reproductive tract.

Experimental design

The manuscript articulates its research question with clarity and is executed with a high degree of rigor and technical proficiency, in alignment with contemporary ethical standards. Nevertheless, it is imperative for the authors to delineate their exclusion criteria explicitly to facilitate the reproducibility of the study. Furthermore, the authors appropriately cite the WHO manual from 2010 within the paper.

Validity of the findings

The manuscript does not appear to be a particularly novel contribution to the literature at this point. The authors should clearly articulate how their work adds value to existing research. While the data presented seems robust and statistically sound, some limitations should have been discussed more thoroughly. For instance, it is unclear how being a single-center study can be considered a significant strength.

In the discussion section, the authors are encouraged to reference recent studies introducing new findings to contextualize their results better. For example, recent research has shown that sperm cells can produce nuclear DNA-based extracellular traps, potentially relying on cell-free DNA. This phenomenon resembles the extracellular traps associated with the systemic inflammatory response related to COVID-19, which may lead to DNA damage following acute infection with SARS-CoV-2. The authors are advised to consult relevant articles to enhance their discussion.

Additional comments

It is important to note that being male is a risk factor for mortality from coronavirus disease. Therefore, it is essential to emphasize in either the conclusion or introduction the necessity of monitoring patients who have experienced severe conditions to assess any damage to their MRT (myocardial reperfusion therapy).

Reviewer 2 ·

Basic reporting

The authors examined the effects of COVID-19 on semen quality before and after the diagnosis of COVID-19.

I think the paper has no novelty and is poorly written with low-quality figures. There are several articles similar to this paper. However, they pointed out that their study is of a single nature, with the exclusion of abnormal infertile patients before infection to better understand the effects of infection on semen analysis, but these reasons are not enough and acceptable.

Moreover, authors have to add more information, such as DNA integrity and fertility rate, which are important in male reproductive health.

My comments are listed below to clarify and improve the manuscript quality.

Keywords
L 35: Please remove Asthenoteratozoospermia.
L 105: Have patients taken antioxidants? What type and at what dosage?
L 181: Please move the reference to the end of the sentence.
L200: Please adjust the format of this reference (25).
L 222: Please check adding multiple references together based on the journal format.

Results:
Please change and improve the quality of the figures.

Experimental design

-

Validity of the findings

Impact and novelty not assessed.

---

## Round 0.2 · Major Revisions

Dear authors,

i apologise, but I require that you submit a new rebuttal with the proper address of the reviewers' questions / comments point by point and not just reporting "it was addressed". In fact, I found the submitted tracked changes document to be rather unremarkable revisions. Whilst this work is already a subject that has been explored in the scientific literature, with articles already published with larger cohorts, the lack of novelty needs to be well justified, while novelty is not a PeerJ requirement, pointless repetition is not in scope. Additionally, DNA integrity and fertility rates data is indeed a now somewhat standard requirement when assessing reproductive health. I also request attention to language and image quality. Thank you in advance.

---

## Round 0.3 · accepted · Accept

Dear authors,

i am happy to let you know that we will now proceed with your manuscript. I am accepting it for publication in PeerJ. Many thanks!

Reviewer 1 ·

Basic reporting

The manuscript titled “The Effects of COVID-19 on Semen Parameter Values in Healthy Males: a single-centre, retrospective study” aims to investigate the impact of SARS-CoV-2 infection on spermatogenesis and its potential effects on patients who exhibited usual semen quality before their COVID-19 diagnosis. Although this topic has been explored in previous scientific literature with larger cohorts, this paper is well-structured and provides valuable insights for practitioners.

Experimental design

The manuscript entitled “The Effects of COVID-19 on Semen Parameter Values in Healthy Males: a single-centre, retrospective study” articulates its research question with clarity and is executed with a high degree of rigor and technical proficiency, in alignment with contemporary ethical standards.

Validity of the findings

The manuscript titled "The Effects of COVID-19 on Semen Parameter Values in Healthy Males: a single-centre, retrospective study" does not appear to be a particularly novel contribution to the literature at this point. The authors now clearly articulated how their work adds value to existing research. The data presented seems robust and statistically sound, and some limitations were discussed more thoroughly.

Additional comments

Dear authors,
After carefully addressing all the reviewers' questions and suggestions, I recommend proceeding with the publication of the article in the journal.